# A High Sensitivity Self-Powered Wind Speed Sensor Based on Triboelectric Nanogenerators (TENGs)

**DOI:** 10.3390/s21092951

**Published:** 2021-04-23

**Authors:** Yangming Liu, Jialin Liu, Lufeng Che

**Affiliations:** College of Information Science & Electronic Engineering, Zhejiang University, Hangzhou 310027, China; yangmingliu@zju.edu.cn (Y.L.); jialinliu@zju.edu.cn (J.L.)

**Keywords:** self-powered sensor, triboelectric nanogenerator, wind speed detection, high sensitivity

## Abstract

Triboelectric nanogenerators (TENGs) have excellent properties in harvesting tiny environmental energy and self-powered sensor systems with extensive application prospects. Here, we report a high sensitivity self-powered wind speed sensor based on triboelectric nanogenerators (TENGs). The sensor consists of the upper and lower two identical TENGs. The output electrical signal of each TENG can be used to detect wind speed so that we can make sure that the measurement is correct by two TENGs. We study the influence of different geometrical parameters on its sensitivity and then select a set of parameters with a relatively good output electrical signal. The sensitivity of the wind speed sensor with this set of parameters is 1.79 μA/(m/s) under a wind speed range from 15 m/s to 25 m/s. The sensor can light 50 LEDs at the wind speed of 15 m/s. This work not only advances the development of self-powered wind sensor systems but also promotes the application of wind speed sensing.

## 1. Introduction

There is often a large amount of unavailable environmental energy, due to the difficult harvesting in our daily life [1]. To exploit the large amount of energy, researchers continue to explore and develop sustainable energy harvesting strategies such as solar energy, electromagnetic energy, wind energy, vibration energy, and water wave energy [2,3,4,5], which also can be applied to a self-powered sensor. The sensor network technology has attracted much attention [6,7,8,9] with the development of Internet of Things (IoT) and artificial intelligence. However, the traditional power supply for sensor systems is batteries, with disadvantages such as high cost, difficulty to maintain, service life, and environmentally unfriendliness, and so forth [10,11,12,13]. Therefore, self-powered systems have gradually attracted attention.

Wind energy has good factors such as a wide distribution, convenient collection, and independence on weather conditions. It is an ideal sustainable energy source for self-powered sensors [14,15,16,17,18]. The traditional wind sensor mainly relies on the principle of electromagnetic induction and is measured through wind cups. However, the wind sensor based on electromagnetic induction has many disadvantages such as complex structure, safety risk, and high cost, which restricts its application on small self-powered sensors [19,20]. Zhonglin Wang proposed the new energy harvesting concept of triboelectric nanogenerator (TENG) in 2012, which has since attracted a lot of researchers’ attention [21]. Practical methods have attracted much attention in the field of self-power sensors. Compared with traditional battery energy, the self-powered sensors based on TENGs have the outstanding characteristics of low cost, simple preparation process, a wide selection of materials, and high energy conversion efficiency [22,23,24,25,26,27,28,29,30,31,32,33,34,35,36,37,38,39]. Many excellent self-powered sensors based on TENGs, such as acceleration sensors, vibration sensors, biosensors, and wind sensors have been reported in recent years [22,23,27].

In this article, we propose a self-powered wind speed sensor based on TENGs with a trapezoidal structure. The self-powered wind sensor is based on the frictional electrification and electrostatic induction coupling between two parallel polytetrafluoroethylene (PTFE) films sputtered with metal electrode layers and an Al/Kapton/Al film to achieve power generation. Each sensor contains two identical TENGs, each TENG can be used to measure wind speed to make sure of the correctness of the measurement. We study the influence of the trapezoidal structural parameters on the output performance of the wind speed sensor. Then, we select a set of parameters of the sensor, the sensitivity of the wind speed sensor is 1.79 μA/(m/s) under the wind speed range from 15 m/s to 25 m/s. This work promotes the potential application of TENGs in wind energy harvesting and related self-powered sensor systems.

## 2. Experimental Section

### 2.1. Fabrication of the TENG

The acrylic plates of different sizes are cut by a laser or a special acrylic cutter, including two trapezoids and two rectangles. The trapezoids have a topline of 25 mm, baseline of 45 mm, and height of 80 mm, while the rectangles’ have a length of 85 mm and a width of 12 mm. The PTFE film is sputtered with a metal Al electrode on one side, and the PTFE film is cut into a trapezoidal structure with a topline of 20 mm, baseline of 40 mm, and height of 80 mm. We fix the Al side of the Al/PTEF film to the trapezoidal acrylic plate by packing tapes. Electrode wires connect the electrode layer with the external circuit. The four acrylic plates are fixed with special glue to form an acrylic tube. The Al/Kapton/Al film is cut into a trapezoid shape. Its structural parameters are the same as the PTFE film. Then, we fix the Al/Kapton/Al film on the inner area of the acrylic tube. From this, we get a wind speed sensor.

### 2.2. Measurement Set Up

For the measurement of TENG, the wind was supplied by a blower, and the wind speed was measured by an anemometer (Victor 816A, Five Long Automation Equipment Co, Shanghai, China.). Besides, all electrical signal measurements were carried out through the programmable Labview platform, which consists of Keithley 6514 and DAQ (data acquisition) modules.

## 3. Results and Discussion

### 3.1. Construction of the Wind Speed Sensor

Figure 1a shows the schematic diagram of the wind speed sensor. The self-powered wind speed sensor consists of two upper and lower layers of PTFE film sputtered with Al metal electrodes and a layer of Al/Kapton/Al in the middle. To improve the electrical characteristics of the electrode layer, we sputter 20 nm Ti film on the Kapton film and the PTFE film, and then we sputter the Al film, which can make the aluminum film more difficult to fall off. The metal Al film sputtered on both sides of the Kapton film is used as a friction layer and an electrode layer simultaneously. The overall dimensions of the friction layer part of the device are a trapezoidal membrane: topline ~20 mm, baseline ~40 mm, and height ~80 mm. The distance between the top PTFE film and the bottom PTFE film is 12 mm. The topline and baseline of the acrylic shell are 5 mm longer than the film to prevent scratches between the film and the side shell when the film vibrates. Figure 1b shows the two TENGs of the sensor. Each TENG concludes a PTFE/Al film and an Al film, both of the two TENGs can be used to measure wind speed so that we can make sure that the measurement is correct by two measurements. Figure 1c shows the top view of the fabricated wind speed sensor, and Figure 1d–f shows the dimension of the wind speed sensor. The trapezoid has a topline of 25 mm, a baseline of 45 mm, and a height of 80 mm. In Figure 1d,e the trapezoidal structure including h_r_ is the length of the topline of the Al/Kapton/Al film, which is 20 mm; h_b_ is the length of the baseline of the Al/Kapton/Al film, which is 40 mm; H_r_ is the length of the topline of the Al/PTFE film, which is 25 mm; H_b_ is the length of the baseline of the Al/PTFE film, which is 45 mm. And the thickness (D_PTFE_) of the sensor is 12 mm. We present more physical images in Appendix A in the supporting information.

### 3.2. Operating Principle of the Wind Sensor and Simulation

Figure 2a shows the working principle of the wind speed sensor. The sensor contains two TENGs. TENG1 is composed of the top layer of PTFE film and the Al on the upper layer of the Al/Kapton/Al film. TENG2 is composed of the bottom ones. In the initial state, the Al/Kapton/Al film has no frictional contact with any PTFE film with no charge transfer between the two TENGs (Figure 2a-i). The Al/Kapton/Al film is in frictional contact with the PTFE film on the top layer under the action of the wind-induced vibration. Negative charges will be generated on the PTFE surface and positive charges will be induced on the surface of Al [35,36]. The generated charge of opposite polarity is completely balanced as shown in Figure 2a-ii. The negative charge produced can be retained on the surface of PTFE film, due to its insulator. Al/Kapton/Al film and the top PTFE film begin to separate as the wind is applied to the sensor continuously. These triboelectric charges cannot be compensated, resulting in a potential difference between the two electrodes and driving electrons from the top electrode to the middle one (Figure 2a-iii). Until the Al/Kapton/Al film contacts with the bottom PTFE film, negative charges accumulate on the PTFE film of the bottom layer. Meanwhile, positive charges accumulate on the Al film to reach the equilibrium state again (Figure 2a-iv). When the Al/Kapton/Al film separates from the bottom PTFE film and moves upwards, the electrical signal will be generated again in the external circuit. Similarly, the electrical signal also will be generated again in the external circuit as the Al/Kapton/Al film moves from top to bottom. Accompanying the Al/Kapton/Al film moving up and down, the two TENGs inside will generate a flow of electrons back and forth, generating an AC signal in the external circuit. To verify the working principle, the motion state of the film is simplified and calculated by the COMSOL Multiphysics 5.5 software. As shown in Figure 2b, the middle Al/Kapton/Al film is in the middle of the distance. When the positions are at 5 mm, 2 mm, −5 mm, the surface potential distribution of the upper and lower TENGs can be seen by simulation. For the upper TENG1, when the middle Al/Kapton/Al film is closer to the bottom PTFE, the greater the potential difference between the top electrode to the middle one. For the lower TENG2, when the middle Al/Kapton/Al film is approximately closer to the top PTFE film, the greater the potential difference between the bottom electrode to the middle one. Therefore, the above simulation results are consistent with the working principle described in Figure 2a.

### 3.3. Output Performance of the Sensor

We test the electrical output performance of the design under the wind speed range from 15 m/s to 25 m/s. Figure 3a,c show the electrical output performance of TENG1 of the device. Figure 3b,d shows the electrical output performance of TENG2. It can be seen from Figure 3a,b that the output performance of TENG1 and TENG2 are roughly equal under the same external conditions which attribute to the symmetrical structure of the two TENGs. Each TENG can be used for wind speed measurement. Figure 3c,d is the trend of the rectified short-circuit current of TENG1 and TENG2 with the wind speed. It can be seen from Figure 3c,d that the peak average current of Wind-TENG varies with the wind speed increases sharply within the flow velocity range from 15 m/s to 25 m/s. The linear relationship between the wind speed and the current can be obtained by linear fitting, and the sensitivity of wind speed is about 1.79 μA/(m/s).

The output performance of the energy harvesting device is usually related to its geometric size. Therefore, we study the influence of different geometrical parameters on its sensitivity. From Figure 4a, we can see that the rectified short-circuit current varies with the distance between PTFE. It can be seen that with the increase of the distance between PTFE films, the rectified short-circuit current increases first and then decreases. This trend can be explained by the short-circuit transfer charge formula of the contact-mode TENG [37]:(1)$$Q_{SC}=\frac{S\sigma x}{d_{0}+x}\;$$
where “S” is the contact area, “*σ*” is the surface charge density, “x” is the distance between the friction layers, and “$d_{0}$” is the thickness of the dielectric layer. It can be seen that the rectified short-circuit current is restricted by the friction contact area and the distance between the two friction layers in the TENG. When the wind speed is constant, the amplitude of the Al/Kapton/Al film is also constant. It means the contact area will decrease with the increase of the distance between the upper and lower PTFE films. Furthermore, the decrease of the contact area will make the charges decrease. However, seen from the formula above, as the distance between the PTFE films increase, the QSC will increase. Therefore, the two impacts of distance cause the variation trend of the output current. From Figure 4b, we can see the variation trend of the wind speed sensor rectified short-circuit current with the length of the Al/Kapton/Al film. It can be seen that the rectified short-circuit current output is significantly reduced when the length is 90 mm or higher. When the film length is long, due to the relationship between the size and weight of the Al/Kapton/Al film, the fluttering motion becomes chaotic and an irregular spanwise direction bending of the flutter motion can be observed, which leads the film to a double contact behavior. This behavior will reduce the contact area and reduce its output [33]. Besides, we also discuss other geometric dimensions such as the width of the trapezoidal Al/Kapton/Al film (the length of the upper and lower bases). Since the flutter frequency of the film depends to a certain extent on the mass ratio of the material itself [34], the flutter frequency of the Al/Kapton/Al film will be affected and the output will be affected when the weight of the strip is too large. It can be seen that the rectified short-circuit current increases significantly when the width of the topline of the film is increased from 10 mm to 20 mm, however, the width of the topline of the film is increased from 20 mm to 25 mm. Although the area of the film increases, the increase of friction area act on the output current is not obvious, which is caused by the mass ratio of the film material itself [34]. To further study the influence of mass ratio on output current, we conducted an experimental analysis on the ratio of the topline length and baseline length of the trapezoidal Al/Kapton/Al film, as shown in Figure 4d. The output is the highest when the ratio of the length of the topline and the length of the baseline of the film is 1:2. However, when the ratio of the length of the topline and the length of the baseline of the film is relatively large, the dither frequency of the film decreases due to the mass ratio of the film itself resulting in a decrease in output. We supply more information about vibration frequency in Appendix A in the Supporting Information. We can see that with the width of Al/Kapton/Al film increase, the vibration frequency of the sensor decreases, as shown in Appendix A. The decrease of vibration frequency will cause the current decrease. However, with the width of Al/Kapton/Al film increasing, the contact area will increase, which means more transfer charges. Therefore, the two impacts of width cause the variation trend of output current in the manuscript. This explanation also applies to the influence of the ratio of the topline and baseline length of the Al/Kapton/Al film.

After the study about the influence of different geometrical parameters on its sensitivity, we determined the sensor geometrical parameters as follows: the PTFE film distance ~12 mm, length ~80 mm, the topline of the trapezoid ~20 mm, and the ratio of the topline and baseline of the trapezoid ~1:2. To calculate the maximum output power of the sensor, we studied its electrical output dependent on load resistance under the range from 10 kΩ to 1 GΩ. As shown in Figure 5a, the output current decreases with the load resistance. Through $P=I^{2}R$, we can obtain the maximum output power of 1.2 mW and the internal resistance of about 5 MΩ.

The wind speed sensor can generate a legible current signal when the Al/Kapton/Al film float up or down. Consequently, a self-powered wind speed sensor can be constructed. The current signal is directly sent to a Labview platform for processing through a data acquisition (DAQ) module. Then, by calculating the average value of the peak current, the computed output current can be displayed on the screen. The DAQ model is shown in Appendix A in the Supporting Information.

Through previous experiments, we can see that within the flow rate range from 15 m/s to 25 m/s, the output current has a smooth linearity. Therefore, the upper and lower TENG can be used together to form a wind speed sensor. The circuit design is shown in Figure 5b. The open-circuit output current of each TENG can be used for wind speed detection. The value of the resistance is related to the wind direction. The previous experiment showed that the relationship between wind speed and current can be obtained linearly: I=k×v+b, where *k* = 1.79 and *b* = −11.4704. “v” is the external wind speed, “I*”* is the output current. For completeness, we made a Labview program that convers the output current into wind speed, as Appendix A shown. The program includes two parts. One of the parts is a peaks acquisition model which can calculate the obtained data and then obtain a computed current value. The other part is a formula calculation model, in which we input a computed current value to obtain the corresponding wind speed value. At the same time, we can see the stability of the wind speed sensor from Figure 5c,d. Figure 5c,d shows a durability test of the wind speed sensor at the wind speed of 25 m/s to manifest the stability and reliability of the wind speed sensor. Besides, the self-powered acceleration sensor has a capacity of lighting up 50 commercial LEDs. When we connect the TENG to the LEDs through the rectifier circuit, the wind speed used in the experiment at 15 m/s, 50 LEDs are lit, as shown in Figure 5e. The self-powered wind speed sensor shows TENG’s broad application prospects in meteorological monitoring and environmental measurement.

## 4. Conclusions

In summary, we propose a high sensitivity self-powered wind speed sensor based on triboelectric nanogenerators. The sensor consists of the upper and lower two identical TENGs, each TENG can detect the wind speed so that we can make sure that the measurement is correct by two measurements. We study the influence of geometric sizes on the rectified short-circuit current of the sensor. We determine the sensor geometrical parameters as follows: the PTFE film’s distance ~12 mm, length ~80 mm, the topline of the trapezoid ~20 mm, and the ratio of the topline and baseline of the trapezoid ~1:2. The output current of the sensor shows a good linear change with wind speed under the wind speed range from 15 m/s to 25 m/s, which achieves a sensitivity of 1.79 μA/(m/s). Its maximum output power can be reached 1.2 mW at the wind speed of 25 m/s. Furthermore, we have demonstrated the good self-powered capability of the device by lighting 50 LEDs at the wind speed of 15 m/s. This work not only promotes the development of self-powered wind sensors but also promotes the application of TENG in wind speed measurement.

## Figures and Tables

**Figure 1 sensors-21-02951-f001:**
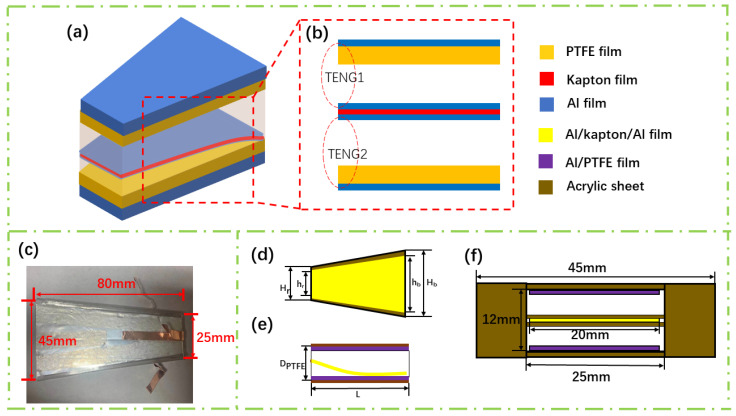
(**a**) Schematic diagram of the wind speed sensor. (**b**) Dual TENG (TENG1 and TENG2) structure of the wind speed sensor. (**c**) The physical image of the wind speed sensor. (**d**) The top view of the wind speed sensor. (**e**) The side view of the wind speed sensor. (**f**) Schematic diagram of the air inlet.

**Figure 2 sensors-21-02951-f002:**
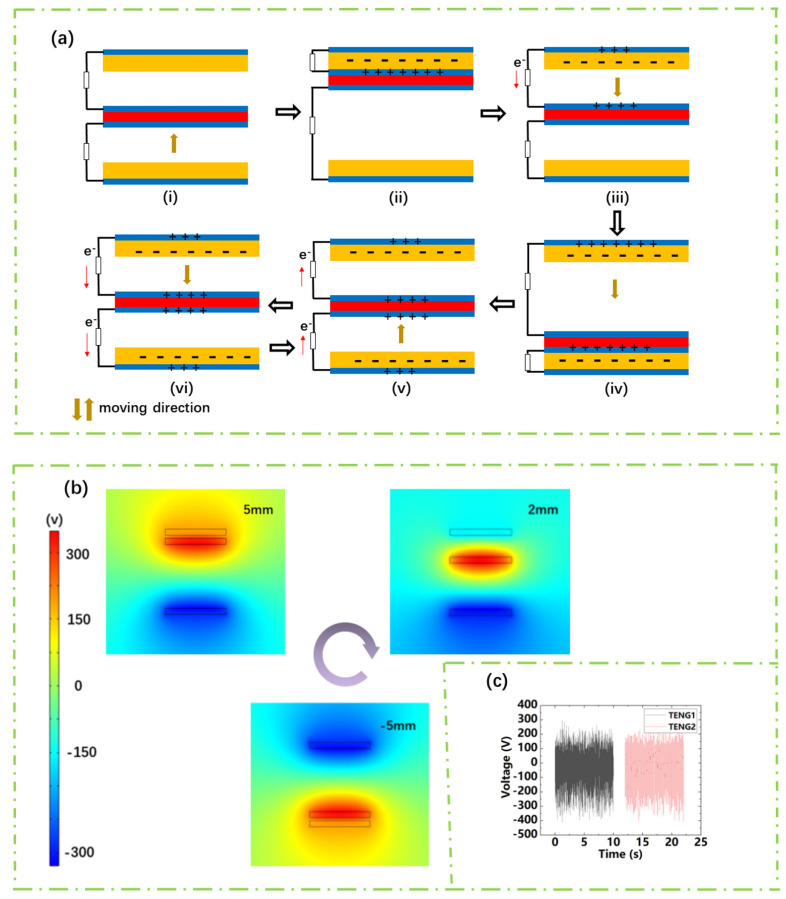
(**a**) Wind-TENG working principle. (**b**) Wind-TENG COMSOL simulation results. (**c**) TENGs open-circuit voltage.

**Figure 3 sensors-21-02951-f003:**
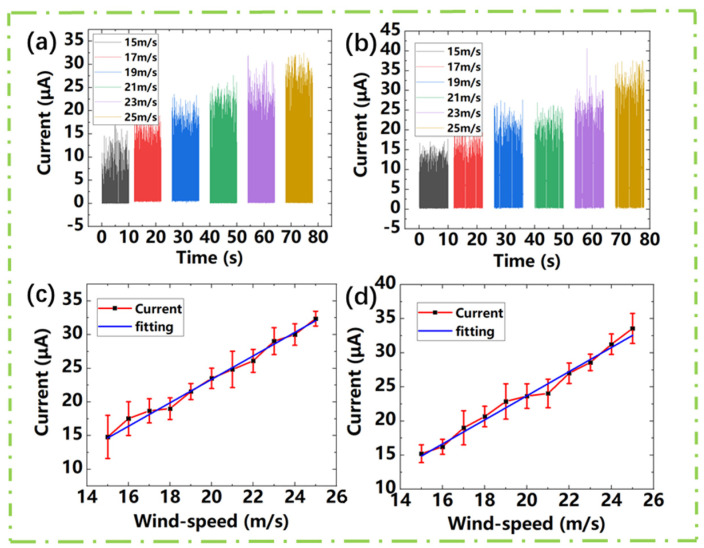
Wind-TENG output curve changes under the wind speed from 15 m/s to 25 m/s. (**a**) TENG1 rectified short-circuit current. (**b**) TENG2 rectified short-circuit current. (**c**) TENG1 rectified short-circuit current changes with wind speed. (**d**) TENG2 rectified short-circuit current changes with wind speed.

**Figure 4 sensors-21-02951-f004:**
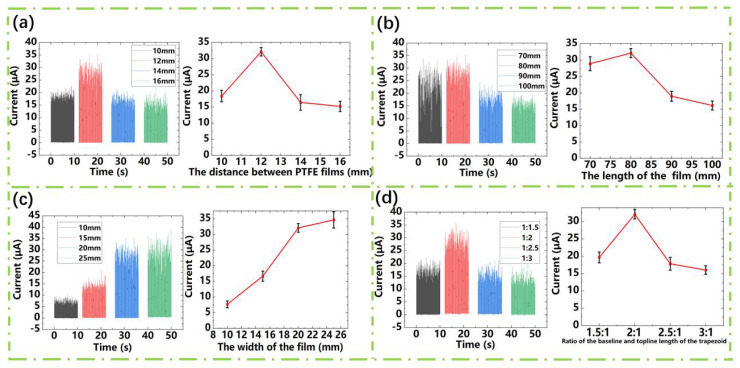
The influence of geometric size on the rectified short-circuit current. (**a**) Current vs. distance between the PTFE friction layers. (**b**) Current vs. the Al/Kapton/Al film length. (**c**) Current vs. the Al/Kapton/Al width of the film. (**d**) Current vs. the ratio of the topline and baseline length of the Al/Kapton/Al film.

**Figure 5 sensors-21-02951-f005:**
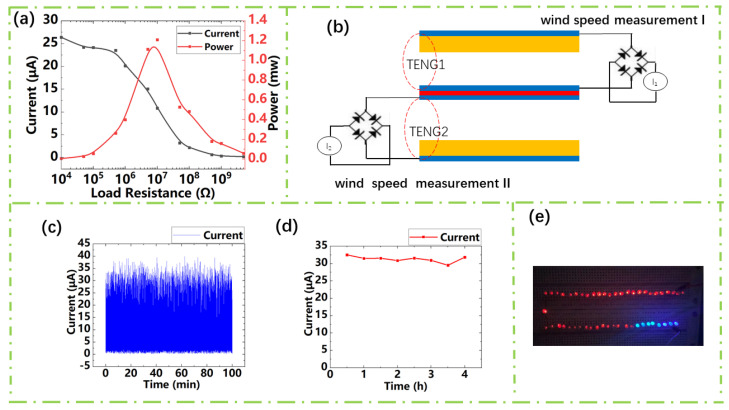
(**a**) Dependence of TENG output current/power on external resistance. (**b**) Circuit design diagram of a sensor for speed measurement. (**c**) The current output of the wind speed sensor lasts for 100 min under the wind speed of 25 m/s. (**d**) The current output of the wind speed sensor lasts for 4 h under the wind speed of 25 m/s. (**e**) The actual picture of TENG powering the LEDs.

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
