# Peer review of "A High Sensitivity Self-Powered Wind Speed Sensor Based on Triboelectric Nanogenerators (TENGs)"

_sensors, 2021, doi:10.3390/s21092951_

Round 1

Reviewer 1 Report

  1. It is mentioned that the acrylic plates are cut into trapezoids in line 59. What are the advantages of this structure?
  2. It is mentioned that the sensor contains two identical TENGs in line 49. Why does the Kapton film bend and connect with the lower PTFE film but not with the upper PTFE film in Figure.1(a)?
  3. Figure.1(e) shows that the Al/Kapton/Al film is sandwiched between two acrylic plates, which is not explained in the paper. If so, how does the Al/Kapton/Al film move to produce friction? If not, how is the Al/Kapton/Al film fixed?
  4. Is the ‘PTFE’ miswritten as ‘PTFF’ in line 104?
  5. It is mentioned that the Al/Kapton/Al film is in frictional contact with the PTFE film on the top layer under the action of the wind-induced vibration.in line 106. Please draw a picture to illustrate how the wind moves the Al/Kapton/Al film.
  6. Figure.2(c) shows that the open circuit voltage reaches 300V. However, the output voltage is 3.3V in reference 21. What optimizations have been made to increase the output voltage greatly?

Author Response

Dear Reviewer 1,

Thanks for your comments and suggestions. We have made changes and provide a point-by-point response. Please see the attachment.

Reviewer 2 Report

In this work the authors study a flutter type TENG for wind energy harvesting. They experimentally optimize the device structure. The novelty of the device is only that it has trapezoidal shaped film. Here are the reviewers comments for the paper:

  1. Minor corrections:
    1. On line 60 the authors say: "while rectangles’ length of 85 mm and width of 12 mm." Please indicate this rectangle in Figure 1.
    2. On line 73 the authors mention the use of DAQ. Please mention the DAQ model.
    3. Heading of Section 3.1 should be changed to "Construction of the wind speed sensor" or something similar.
    4. In Figure 1d, the Hr,Hb are for Al/PTFE film but are incorrectly marked brown color.
    5. In Figure 3 and 4 the current output seems to be rectified (not AC type). This should be mentioned in the manuscript.
    6. Equation 1 reference on line 157 should be [37] not [36].
    7. Figure 4d caption is incomplete "top and bottom length…"
    8. Figure 5b mentions "wind direction measurement" but this not described in detail in the manuscript.
  2. The structure discussed in the paper has vibration modal frequencies as discussed in references "S. Wang et al. Advanced Materials 27 (2015) 240-248." and "W. Kim et al. Nano Energy 56 (2019) 307-321." This could effect the TENG output. On line 184 the authors mention "dither frequency". They should further clarify what they mean by this term. The authors should atleast mention the output frequency at a specific speed as in this recent reference "Y. Xia et al. Micromachines 12 (2021) 366."
  3. What is the wind input side? As in reference "J. Bae et al. Nature communications 5 (2014) 1-9." the input side is wider than the TENG stage since this increases the wind velocity. In a trapezoidal structure like this work the variable fluid mechanics theory should be considered to explain the experimental results in Figure 4.
  4. For completeness the authors should make a Labview program that converts the TENG-1 and TENG-2 output into wind speed indicator based on their derived linear relationship on line 205 (I=k*v+b).

Author Response

Dear Reviewer 2,

Thanks for your comments and suggestions. We have made changes and provide a point-by-point response. Please see the attachment.

Round 2

Reviewer 1 Report

The authors have addressed most of my questions. I recommend this manuscript to be publicated in this journal.

Reviewer 2 Report

The authors have addressed my suggested corrections and made the suggested additions to improve the quality of the manuscript. Therefore i can recommend the revised manuscript for publication as Communication in Sensors journal.